# Dietary Bioactive Fatty Acids as Modulators of Immune Function: Implications on Human Health

**DOI:** 10.3390/nu11122974

**Published:** 2019-12-05

**Authors:** Naren Gajenthra Kumar, Daniel Contaifer, Parthasarathy Madurantakam, Salvatore Carbone, Elvin T. Price, Benjamin Van Tassell, Donald F. Brophy, Dayanjan S. Wijesinghe

**Affiliations:** 1Department of Microbiology and Immunology, School of Medicine, Virginia Commonwealth University, Richmond, VA 23298, USA; ngajenthrakum@mymail.vcu.edu; 2Department of Pharmacotherapy and Outcomes Sciences, School of Pharmacy, Virginia Commonwealth University, Richmond, VA 23298, USA; dcontaifer@vcu.edu (D.C.); etprice@vcu.edu (E.T.P.); bvantassell@vcu.edu (B.V.T.); dbrophy@vcu.edu (D.F.B.); 3Department of General Practice, School of Dentistry, Virginia Commonwealth University, Richmond, VA 23298, USA; madurantakap@vcu.edu; 4Department of Kinesiology & Health Sciences, College of Humanities & Sciences, Virginia Commonwealth University, Richmond, VA 23220, USA; salvatore.carbone@vcuhealth.org; 5VCU Pauley Heart Center, Department of Internal Medicine, Virginia Commonwealth University, Richmond, VA 23298, USA; 6da Vinci Center, Virginia Commonwealth University, Richmond, VA 23220, USA; 7Institute for Structural Biology, Drug Discovery and Development, Virginia Commonwealth University School of Pharmacy, Richmond, VA 23298, USA

**Keywords:** EPA, DHA, FDA regulations, Immune function, toll like receptors, essential fatty acids, non-essential fatty acids, PPAR

## Abstract

Diet is major modifiable risk factor for cardiovascular disease that can influence the immune status of the individual and contribute to persistent low-grade inflammation. In recent years, there has been an increased appreciation of the role of polyunsaturated fatty acids (PUFA) in improving immune function and reduction of systemic inflammation via the modulation of pattern recognition receptors (PRR) on immune cells. Extensive research on the use of bioactive lipids such as eicosapentaenoic acid (EPA) and docosahexaenoic acid (DHA) and their metabolites have illustrated the importance of these pro-resolving lipid mediators in modulating signaling through PRRs. While their mechanism of action, bioavailability in the blood, and their efficacy for clinical use forms an active area of research, they are found widely administered as marine animal-based supplements like fish oil and krill oil to promote health. The focus of this review will be to discuss the effect of these bioactive fatty acids and their metabolites on immune cells and the resulting inflammatory response, with a brief discussion about modern methods for their analysis using mass spectrometry-based methods.

## 1. Introduction

Dietary fatty acids, either by themselves or via their metabolites, have the capacity to influence human health and health outcomes [1]. A detailed dissection of the components of lipids associated with poor cardiovascular health in the past decade has enabled the identification of putative lipid biomarkers predictive of poor cardiovascular health. Lipidomic analysis to study models of dyslipidemia have shown an accumulation of saturated fatty acids and omega-6 fatty acids-associated lipids [2] and are considered to be inflammatory in nature [3]. Increase in disorders like type II diabetes, cardiovascular diseases, and atherosclerosis, which are highly associated with an unhealthy diet, have brought forth the importance of lipid homeostasis in health and disease. Furthermore, with the advent of lipidomics, an increasing grasp on the diversity of lipid species suggests that the relative abundance of lipids influence outcomes [4], rather than the mere presence or absence of a lipid species. The prevailing dogma suggests that an increase in free omega-3 polyunsaturated fatty acids and associated lipids (e.g., omega-3: omega-6 ratio) is known to promote health in humans and is correlated with lower levels of systemic inflammation. Bioactive lipids, specifically polyunsaturated long chain fatty acids, are classified based on their degree of unsaturation, which is insufficient to infer their biological function, and it is important to discuss the ways in which fatty acids play a role in inflammation and immune function.

Studies on the human lipidome, not limited to the classes of phospholipids, cholesterol esters, triacyl glycerol, and fatty acids have been implicated as an area of vital research for diet and lifestyle-associated disorders. Fatty acids differing in their position of desaturation (omega 3 vs. omega 6) play distinct roles in the body and are the primary focus of our discussion (Figure 1). These fatty acids are considered essential fatty acids as humans are unable to synthesize the basic precursors. The most basic dietary precursor for omega-3 fatty acid is the alpha-linolenic (ALA) acid or linolenic acid which is a fatty acid (FA) containing 18 carbons with three double bonds (18:3)with the first double bond from the non-carboxyl end beginning at the third carbon (n-3) and abbreviated in whole as ALA FA 18:3 n-3. This lipid is converted to the anti-inflammatory eicosapentaenoic acid (EPA FA 20:5 n-3), and docosahexaenoic acid (DHA FA 22:6 n-3) (Figure 1). Dietary omega- 6 fatty acids like linoleic acid (FA 18:2 n-6) are converted to gamma-LA (FA 18:3 n-6) and arachidonic acid (AA FA 20:4 n-6), and have distinct roles in inflammation (Figure 1). These fatty acids serve as precursors to many bioactive lipids. When taken via diet, they are converted to monoglycerides and free fatty acids in the intestinal lumen, followed by incorporation into chylomicrons and lipoproteins for circulation within the bloodstream. Omega-3 fatty acids are anti-inflammatory, whereas omega-6 fatty acids are pro-inflammatory, and this association depends on the lipid metabolites produced downstream from these precursors. Biochemically, higher concentrations of dietary bioactive lipids like EPA and DHA compete with AA for synthesis of lipid mediators and can tip the balance towards less inflammatory/pro-resolution phenotypes [5,6,7]. Resolution may occur when the conversion of arachidonic acid to inflammatory mediators by cyclooxygenase-2 (COX-2) is competed off by EPA and DHA to produce pro-resolution lipids (reviewed in [6]). In addition to their metabolic flux, these fatty acids are known to competitively modulate signaling through pattern recognition receptors and G protein coupled receptors (GPR40) [7,8] on leukocytes [9,10,11] and thus reduce the risk of inflammation-mediated cardiovascular disease progression. Metabolites of long chain fatty acids, also known as eicosanoids, can interact with G-protein-coupled receptors GPCRs [8] and have been implicated in the development of atherosclerosis. Thus it may be possible to ultimately allow for targeted, personalized applications of lipid formulations for managing systemic inflammation perpetrated by particular cell types of the immune system (T-cells, B cells, and dendritic cells) and the treatment of disorders associated with unhealthy diet [12,13]. While this is an exciting area of research, the narrow window of physiological response demands accurate quantitation of lipid species which is now possible with advances in the field of lipidomics.

Advances in liquid chromatography coupled to high resolution mass spectrometry, and more recently ion mobility methods [14], have enabled comprehensive characterization of the mammalian lipidome. Chromatographic and ion mobility separation prior to mass spectrometry is required to overcome the challenge of isobaric overlap when analyzing lipids by mass spectrometry. In this regard, reverse phase chromatographic methods enable the separation of lipids based on hydrophobicity (Figure 3b) primarily resulting from the fatty acid composition of the lipid species. On the other hand, hydrophilic interaction liquid chromatography (HILIC) [15] provides class-based separation of lipids based on the lipid molecule head groups, the primary determinants of lipid classes. The use of such complimentary methods has enabled the quantification of thousands of lipid species which can be used for targeted monitoring of lipids for personalized therapeutic approaches [12,13]. While liquid chromatography-based separation suffices for most lipid separations, to date, separation in the gas phase (gas chromatography) prior to mass spectrometric analysis remains the most reliable method free fatty acid analysis. It should be noted that newer separation methods like supercritical fluid chromatography coupled to mass spectrometry is also demonstrating great promise towards quantitative analysis of bioactive lipids.

In the following review, we seek to summarize the role of lipids in immunopathology while discussing the relevant advances in analytical and statistical methods in lipidomic research for studying a broad variety of inflammation-associated diseases.

## 2. Fatty Acids and Immune Function

### 2.1. Fatty Acids Influence Inflammatory Repertoire

Fatty acids are classified as short chain, medium chain, and long chain fatty acids. There are three primary means by which fatty acids can influence the inflammatory repertoire of the host; substrates for biosynthesis of inflammatory mediators, activation of cell receptors [16,17], and modulation of membrane fluidity to alter cell function. Bulk of the work in this field has focused around the mechanisms by which polyunsaturated fatty acids (PUFA) act as substrates for the biosynthesis of inflammatory eicosanoid mediators (Figure 1). While some work has focused on delineating the mechanism by which fatty acids interact with cell surface receptors or even modulate cellular function through the production of oxylipins [18], its implications on overall health, and their efficacy as interventions remain elusive. Dietary intake of omega-3 unsaturated fatty acids provides precursors for the production of anti-inflammatory lipids like five series leukotrienes (LTx_5_) or three series prostanoids (PGx_3_). On the other hand, an increase in omega-6 fatty acids leads to the production of pro-inflammatory mediators like four series leukotrienes(LTA_4_, LTB_4_, LTC_4_, LTD_4_, LTE_44_) and two series prostanoids (PGx_2_) [19] (Figure 1). The metabolites produced, further regulate inflammation by feedback inhibition of biosynthetic enzymes [7,20]. This autocrine mechanism of regulation by the two series of prostaglandins has been well studied [3,21]. In the body, fatty acids and their metabolites [22] are usually present together and influence the end result by their relative concentration in the milieu. The initial inflammatory response, as represented by the production of LTx_4_, is important for the infiltration of neutrophils to the site of infection and thereby beginning the cascade for the production of pro-inflammatory cytokines [23]. The incorporation of these bioactive lipids into membrane phospholipids affects membrane fluidity and surface receptor expression and regulates the function of immune cells. Enrichment of T cells and neutrophils with omega-3/omega-6 fatty acid supplemented in the media provides an evidence for the incorporation of exogenous fatty acids into membrane phospholipids of immune effector cells [24,25]. Taken together it suggests that immune cells are capable of incorporating exogenous fatty acids into the cell membrane. The incorporation of fatty acids changes the membrane architecture and signaling, thereby altering the function of cell surface pattern recognition receptors [26,27].

Reports investigating this phenomenon have provided some insights into the incorporation of fatty acids into the cell membrane of macrophages [28]. This effect is seen specifically in activated polymorphonuclear neutrophils (PMN’s) where there is a loss of membrane incorporated unsaturated fatty acids resulting from heightened intracellular phospholipase activity (cPLA_2_) due to the activation of leukocytes [29,30] downstream of cell surface receptors like Toll-like receptors (TLRs).

Lipids enriched in dietary fatty acids can influence the inflammatory profile during episodes of sterile or infectious inflammation. Enrichment of omega-3 fatty acids in the media or diet improves the function of lymphocytes by improving mitogen-mediated activation of immune cells [31,32,33]. An omega-3 rich diet further promotes the development of a T_H_2-type immune response [34,35] promoting the production of associated anti-inflammatory cytokines like IL-4 [34,35], and reduction of pro-inflammatory TNF-α [20,27,32,33,34]. In contrast, a coconut oil-based diet (rich in saturated fatty acids) led to the development of an IFN-γ dominant cytokine profile, characteristic of a T_H_1 immune response. While a T_H_1 response may be beneficial in a parasitic infection, a T_H_2 response is preferred in most cases. Expanding on the effect of omega-3 PUFA on T cell subtypes, it is interesting to note that not only a fish oil-enriched diet, but also purified EPA and DHA suppressed IL-17 production from T_H_17 cells resulting in reduced STAT-3 phosphorylation and ROR-γτ expression. This concomitant decrease in T_H_1/T_H_17 in response to omega-3 fatty acids (EPA and DHA) may have implications on the ability to reduce enteric inflammation [36]. Similarly, a diet rich in monounsaturated fatty acids (MUFA) like oleic acid (Mediterranean diet) has been shown to improve high density lipoprotein (HDL) function [37] and is protective in patients for high risk for cardiovascular disease [37,38,39] (Table 1 and Table 2).

### 2.2. Fatty Acids Influence Immune Functions by Interacting with Cell Surface Receptors on Immune Cells

In an effort to dispel the confusion about the mechanism of action of polyunsaturated fatty acids and find cognate receptor interaction, some landmark studies have determined a few receptors that fatty acids interact with on host cells like the peroxisome proliferation activating receptor (PPAR) and Toll-like receptors (TLRs) [40,41]. Interaction with these receptors results in the activation of signaling cascades activating the transcription of anti-inflammatory cytokines while suppressing the transcription of pro-inflammatory cytokines (Figure 2). Studies investigating the interaction of fatty acids with Toll-like receptors (TLRs), particularly TLR2 and TLR4, on leukocytes have shown that saturated fatty acids (C12:0 and C16:0) cause an increased expression of cyclooxygenase-2 (COX-2) and phosphorylation of ERK (p-ERK) in a MyD88 independent manner [16]. Consistent with reported literature, it was also found that DHA and other n-3 fatty acids caused suppression of COX-2 and p-ERK. This study dispels any contention of the bacterial contaminants or the influence of bovine serum albumin (BSA) in eliciting the activation of TLRs [16]. It was also found that monocytic cells (THP-1) and macrophages (RAW 264.7) had a heightened response to saturated fatty acids under serum-starved conditions which was influenced by the reactive oxygen species (ROS) status of the microenvironment, while EPA and DHA suppressed this response [40]. It must also be noted that administration of high fat diet rich in saturated fats has been shown to increase endotoxin (LPS) production by gram negative bacteria [4] and thus TLR4 activation. Saturated fatty acids also activate an additional number of pro-inflammatory pathways, such as the one related to the intracellular macromolecular complex Nod-like receptor pyrin domain-containing protein (NLRP)-3 inflammasome, primarily responsible for the production of the pro-inflammatory cytokines IL-1β and IL-18 [42,43,44,45]. Conversely, unsaturated fatty acids inhibit such detrimental effects, exerting anti-inflammatory properties.

Another kind of membrane-associated receptor other than TLR is the PPAR-gamma (PPAR-γ). It is trafficked to the cell surface in membrane lipid rafts and influences the inflammatory response in an NF-κB dependent manner. Some disease presentations like atherosclerosis are seen due to a PPAR-γ dependent expression of oxLDL uptake receptors that lead to the formation of foam cells in fatty streaks (plaques) seen in arteries. In most cases, this is a result of downstream activation of a cell surface receptor like PPAR-γ [10,46,47], which initiates the cascade required for the immediate production of inflammatory lipid mediators like prostaglandins and cytokines (i.e., IL-2) prior to the infiltration of the tissue by immune cells. Research on this front has provided information that fatty acids and other eicosanoids, like PGJ_2_, are ligands for PPAR-γ and may increase the transcription of the PPAR-γ [48,49]. However, the suppression of NF-κB-mediated pro-inflammatory responses by EPA and DHA are independent of PPAR-γ [50]. This modulation of the immune response has also been elicited in the requirement of PPAR-γ for the maturation of dendritic cells and the suppression of macrophage-mediated inflammation by n-3 fatty acids [51]. A more responsive counter part of the PPAR family of receptors to n-3 supplementation are PPAR-α and PPAR-δ. The expression of these genes was observed to be upregulated in mice fed with high fat diets and this increase was suppressed when mice were fed with a high fat diet supplemented with DHA and EPA [52,53]. Thus, n-3 fatty acids are antagonistic to pro-inflammatory pathways regulated by TLR (i.e., saturated fatty acids) and PPAR cell surface receptors.

Similar to the interaction of eicosanoids with membrane spanning G protein coupled receptors, long chain fatty acids also interact with G protein coupled receptors on the surface of cells. An upcoming area of research is the interaction of short chain [54,55,56] and long chain fatty acids with yet uncharacterized G protein coupled receptors and their role in modulating inflammation. For a comprehensive overview of the specific interactions of the free fatty acids with their cognate G protein receptors, the readers are directed to some excellent reviews in this emerging field [57,58].

### 2.3. Fatty Acids Influence Lymphocyte Proliferation and Cytokine Profiles

It has been reported as early as 1988 that inhibition of the lipoxygenase pathway of inflammation prevents the differentiation of monocytes to macrophages after supplementation with arachidonic acid (n-6) [31]. A concomitant increase in PC specific phospholipase activity is observed in instances where linoleic acid (n-6) is found esterified in the sn2 position of phospholipids [31]. Depending on the stage of development of the macrophage, enrichment of cells with n-3 fatty acids such as linolenate (FA 18:3 n-3) promotes a rapid acute immune response by a reduced production of TNF-alpha when compared to macrophages enriched with n-6 fatty acids such as linolenic acid [20]. A more recent study reiterates the findings that n-3 fatty acids can stimulate the production of IL-4 while saturated fatty acids promote the production of IFN-γ [34]. In addition to influencing cytokine profiles, n-3 fatty acids also promote the expression of the Mac-1 complex (CD-11b/CD-18) on the surface of neutrophils on the cell surface. While the exact mechanism of action remains unclear to date, the expression of Mac-1 on the surface is not dependent on the metabolic production of pro-inflammatory lipids like PGE_2_ or LTB_4_ [59]. Thus, suggesting that while arachidonic acid metabolites do not play a role in the expression of surface markers and differentiation of monocyte derived cell types, they are important in initiating the cascade responsible for terminal differentiation. Arachidonic acid, an n-6 fatty acid previously shown to be enriched in atherosclerotic plaques, has also been suggested to induce the expression of CD36 and scavenger receptor A [49] on the surface of macrophages. An expression of these receptors promotes prolonged residence of macrophages in bloodstream and the development of foam cells due to increased receptor-mediated oxLDL uptake. A review summarizes the connection between dietary nutrients and immune function [35,60] and we elaborate on some of the mechanisms by which dietary lipids influence immune function at the cellular level.

Fatty acids may also regulate the production of pro-inflammatory cytokines by interacting with the NLRP3 inflammasome and modulating the production of IL-1 family of cytokines (IL-1β and IL-18) [61,62]. Signaling cascade downstream of pattern recognition cell surface receptor (TLR) provides the initial signal for the production IL-1 family of pro-inflammatory cytokines [63]. The maturation of the cytokines is controlled by a multiprotein complex called the inflammasome. Though interesting, it is not surprising that the saturated/n-3 fatty acids play antagonistic roles to n-6 fatty acids in inflammasome function [64,65]. While saturated fatty acids like palmitic acid (C16:0) induce the production of IL-18 and IL-1β in an NLRP3 dependent manner, monounsaturated fatty acid (MUFA) like oleic acid (C18:1) and n-3 PUFA inhibit the production of these cytokines [64] and are involved with the transcriptional repression of NLRP3. Taken together, an increased proportion of saturated and n-6 fatty acids inhibit immune cell development and promotes inflammation. MUFA [38,66] and PUFA (n-3) suppress the production of the pro-inflammatory cytokines by interacting with the NLRP3 inflammasome and modulate inflammation at the transcriptional and translational level by suppressing gene expression of the components of the inflammasome and preventing the NLRP3 dependent maturation of IL-1 family of cytokines [43,52,53,62,64,65].

A resurgence of interest in bioactive lipids has been complemented with the characterization of the bioactive lipids composition of M1/M2 polarized macrophages [67]. Studies have shown that the lipidomic composition and the resulting phospholipase subtype mediated activity to influence the development of proinflammatory subtype (M1) over the anti-inflammatory subtype (M2) [67]. While it continues to be determined whether the product of 12-Lipoxygenase (12-HETE) is important for shifting the polarization of macrophages to the M2 subtypes, more work focused on the eicosanoid signaling and its relation to cell types differentiation is required. Resulting from the distinct mechanisms of differentiation to macrophage subtypes, M1/M2 cells have distinct eicosanoid compositions where M1 cells are abundant in LTB4 and PGE2 while M2 macrophages are abundant in pro-resolving mediators from the 5-lipoxygenase pathway (5-LOX) and eicosanoids derived from n-3 fatty acids like EPA and DHA such as resolvin D2 and D5 (RvD2, 5). Even more interesting is that M2 cells produced a higher concentration of PGD_2_, the anti-inflammatory metabolite of arachidonic acid [68,69]. Taken together, these reports suggest an overlooked role of bioactive lipids and their mediators in modulating the outcomes of infection and inflammation. Further emphasizing the need for the analysis of localized lipid metabolism at the sight of interest instead of classical systemic evaluation of efficacy of the efficacy of bioactive lipids in health and disease. 

### 2.4. Maintaining Data Quality and Rigor in Studies Involving Bioactive Lipids

Activity of bioactive lipids is controlled by enzymatic conversion to inactive metabolites or sequestering them in phospholipids and thereby producing oxidized phospholipids. The levels of bioactive lipids in human plasma have been quantified down to the picomolar (pM) range [75], and a reliable way of analyzing modulation in these bioactive lipids is through mass spectrometry-based approaches. For those interested in oxidized phospholipids (i.e., lipids with bioactive lipids esterified to the phospholipid backbone) and their functions, a combination of chromatographic methods and ion mobility methods are required to distinguish phospholipids with traditional fatty acids from phospholipids containing sequestered eicosanoids (Figure 3c).

Prior to determining the abundance and function of phospholipids that contribute to the production of bioactive lipids, the overall lipidomic composition needs to be determined. Infusion-based lipidomics-based approaches such as MS/MS^All^ provide a rapid method to determine lipidomic compositions (Figure 3a). This simplified approach provided a comprehensive coverage of lipids in a short run time [76], enabling the identification of the lipids as well as their constituent components such as the fatty acids. Problems of isobaric overlap can be addressed by liquid chromatography-based tandem mass spectrometry but require significantly longer run times with the advantage of resolving isomeric lipid species, i.e., lipids with similar exact mass, based on their retention time and fragmentation pattern (Figure 3b). Identification and analysis of oxidized phospholipids benefit from the use of liquid chromatography-based separation to determine the lipid classes that sequester the bioactive lipid, followed by the comprehensive characterization of the lipid molecule based on accurate mass and fragmentation by tandem MS (Figure 3c).

With respect to studying omega-3 and omega-6 fatty acids involvement in a disease state or changes occurring at cellular level, comprehensive and targeted LC-MS/MS MRM methods enable simultaneous quantification of more than 100 lipid metabolites, including prostaglandins, leukotrienes, and other eicosanoids, resolvins, protectins, and other free fatty acids like arachidonic acid, eicosapentaenoic acid and docosahexaenoic acid [12,77,78] (Figure 3c (Targeted method)).

## 3. Implications of Bioactive Fatty Acids and Their Metabolites on Human Health

The US Food and Drug Administration (FDA) regulates the requirements for appropriate food labeling in the Title 21CRF101, including labeling of dietary supplements. In 2003, a final rule published in the Federal Register required that trans fatty acids be included in nutrition labeling, based on requests from the Center for Science in the Public Interest, due to the detrimental effects of such nutrients on plasmatic lipoproteins with potential increased risk of cardiovascular and metabolic diseases (i.e., increased low density lipoprotein-cholesterol (LDL-C)) [79,80].

In an effort to regulate the industry and inform the public on food and nutrition, the Food and Agriculture Organization of the United Nations (FAO) and the World Health Organization (WHO) provided a report of an expert consultation on fats and fatty acids in human nutrition. Published in 2008, this guidance is a useful reference on nutritional requirements and recommended dietary lipid intakes. This expert consultation focused on the role of specific fatty acid groups, such as the role of long-chain polyunsaturated fatty acids (LCPUFA) in neonatal and infant mental development, besides their role in maintenance of long-term health and prevention of specific chronic diseases. It was recommended that the n-3 PUFA and n-6 PUFA include more than one fatty acid with distinct attributes and biological function, and labelling them based on category may not be the best path forward. Due to an increasing acknowledgement of the importance of lipids during the initial years of life, this report also describes the requirements and the recommendations for fat and fatty acid for infants of 0–2 years and children of 2–18 years old, respectively [80].

A landmark randomized controlled trial was initially published in 2013 and then republished in 2018, is the ‘Primary Prevention of Cardiovascular Disease with a Mediterranean Diet’ [39,81] (PREDIMED). In the PREDIMED study, the consumption of energy unrestricted high-fat diets supplemented with extra-virgin olive oil or nuts, foods rich in unsaturated fatty acids, induced an impressive 30% relative risk reduction of major cardiac events compared to subjects who were counseled to follow a relatively low-fat diet. In addition, the subjects randomized to the high-fat diets experience a lower incidence of type 2 diabetes, therefore suggesting that the consumption of healthy fats also improves metabolic outcomes, in addition to cardiovascular outcomes [39]. These results not only support the concept that increasing the consumption of healthy fats (i.e., unsaturated n-3 fatty acids) may prevent cardiovascular diseases, but also that perhaps the prior cut-off of 30%–35% of total calories deriving from fat recommended by the FAO and WHO was still a sub-optimal amount, with the risk of precluding healthy subjects as well as patients with established diseases from the beneficial effects of the high-healthy fats diet. In fact, a recent presidential advisory from the American Heart Association [38] emphasized focusing on replacement of saturated fatty acids with unsaturated fatty acids to reduce the risk of cardiovascular diseases, without recommending a specific goal for total fat intake in terms of total % of calories. Other studies assessing the benefit of using polyunsaturated fatty acids in conditions other than cardiovascular disease can be found as follows in (Table 2). While discussions on the efficacy of the use of bioactive lipids in each of these studies is out of the scope of this review, it can be appreciated that sustained supplementation with high amounts of n-3 polyunsaturated fatty acids promote improved health outcomes.

## 4. Conclusions

Diet plays a major role in affecting metabolic and cardiovascular health. Particularly, fatty acids such as omega-3 and omega-6 have been subject of intense scrutiny in the last decades. Multiple studies agree on the fact that supplementation with bioactive lipids results in an increased bioavailability; cell type specific and systemic. Acknowledgement and appreciation of the antagonistic roles of n-3 PUFA/MUFA and SFA/n-6 PUFA in inflammation provide avenues for further research into their mechanism of action. While it has been difficult to determine if fatty acids interact with pattern recognition receptors by cognate interactions, there is consensus about their mechanism of action and final targets (reviewed in [90]). Commonality between studies that report on the benefits of n-3 fatty acids on health is that they are long term studies with high doses of supplementation. Thus, suggesting the need for a sustained intake to modify host lipid composition and thereby reduce the severity of diseases involving lipid-mediated signaling and inflammatory cascade. In addition to that, separate reports suggesting that n-3 fatty acids when present in concentration in excess of 40 uM in vitro and in vivo elicit beneficial effects by suppressing pro-inflammatory responses even at the cellular level argue in favor of determining endogenous concentrations of n-3 fatty acids in studies that did not show an affect with supplementation. In addition to their small window of bioactivity and rapid turnover, the field of study is marred by looking at systemic effects in diseases with localized etiology.

Much like immune function, bioactive lipids classically act in localized autocrine and paracrine circuits, the downstream effects of which determine physiological outcomes. Thus, while systemic evaluation of levels of bioactive lipids are informative of circulating levels, they represent the resting physiological levels unless sampling is performed in a state of active inflammation or disease. Furthermore, in supplementation studies, the quantification of a few bioactive lipids outside of the context of its precursors and metabolites provides incomplete information about their efficacy and may be thought to contribute to the variability in studies involving bioactive lipids.

Further research on testing the efficacy of lipid formulations rich in n-3 fatty acids/MUFAs by advanced lipidomics approaches to specifically monitor the flux of bioactive lipids [91] in relation to inflammation in intervention studies are warranted. Such studies hold promise to dispel any ambiguity that may remain with regards to the role of n-3 fatty acids/MUFAs as beneficial interventions in acute and chronic conditions that lead to the presentation of metabolism-associated disorders like cardiovascular diseases.

## Figures and Tables

**Figure 1 nutrients-11-02974-f001:**
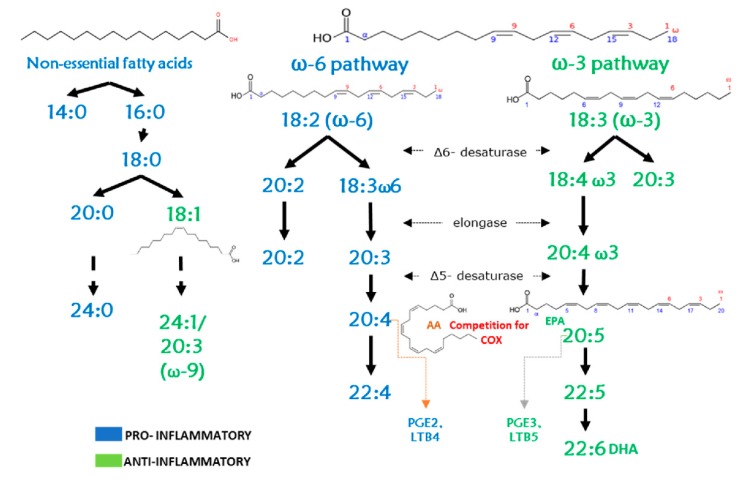
Fatty acid configuration and their role in inflammation-related pathways. n- omega- position of unsaturation, fatty acid nomenclature XX:Y; XX—number of carbon, Y—number of unsaturated bonds. AA—arachidonic acid, EPA—eicosapentaenoic acid, DHA—docosahexaenoic acid, COX—cyclooxygenase, LTB—leukotriene, PGE—prostaglandin.

**Figure 2 nutrients-11-02974-f002:**
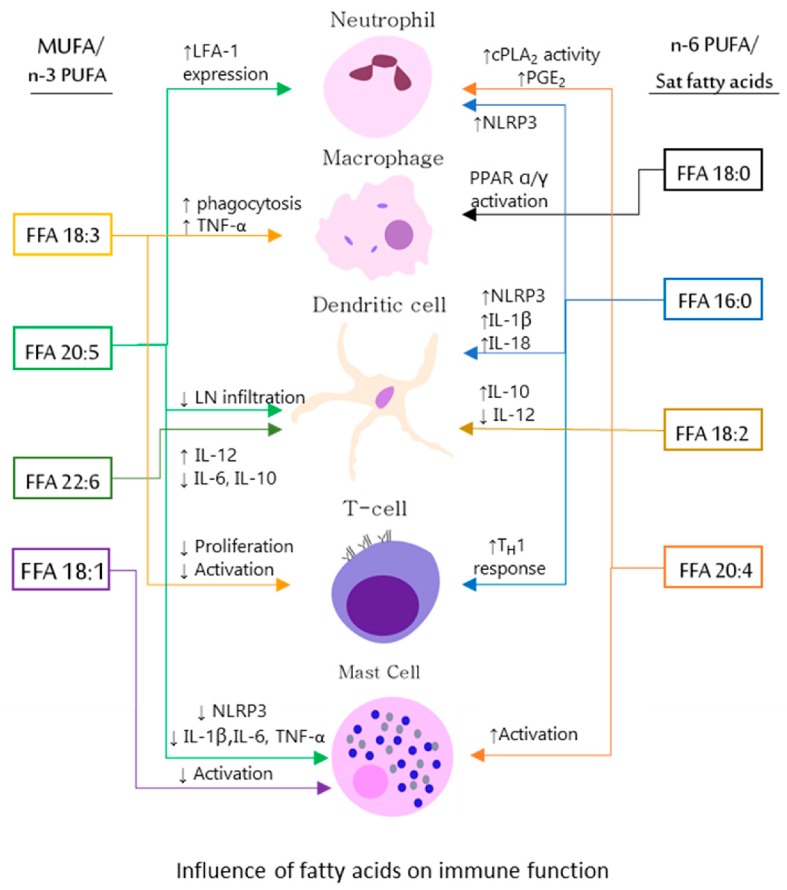
Influence of fatty acids on immune cells as summarized in Table 1. Abbreviations; LFA-1—leukocyte factor antigen-1, IL—interleukin, Th1—T helper type I, TNF-α—tumor necrosis factor alpha, LN—lymph node, cPLA_2_—cytosolic phospholipase A2, PPAR—peroxisome proliferator-activated receptor, NLRP3—nucleotide-binding and oligomerization domain-like receptor, leucine-rich repeat and pyrin domain–containing 3, ↑—levels increase, ↓—levels decrease.

**Figure 3 nutrients-11-02974-f003:**
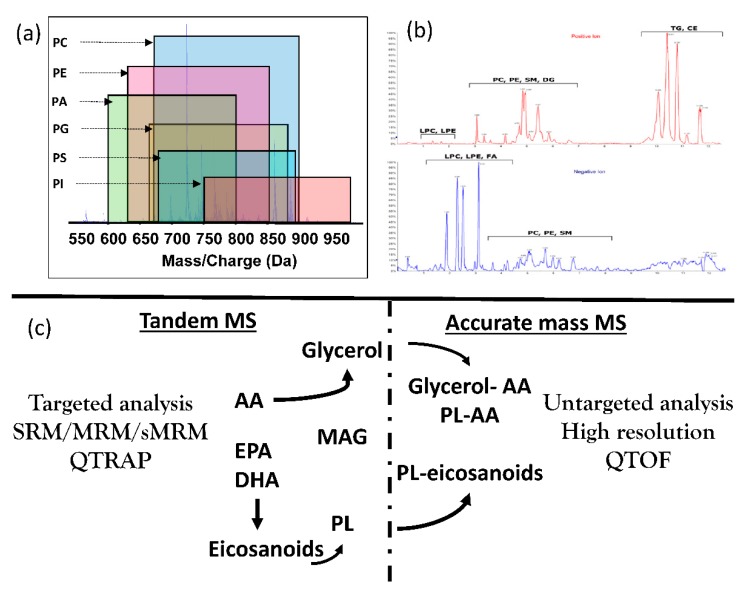
(**a**) Isobaric overlap (kindly provided by Dr. Paul Baker) typically seen while analyzing lipids. (**b**) Separation of lipids based on class in reverse phase chromatography in positive and negative mode. (**c**) Schematic of fatty acid and eicosanoid metabolism and suggested approaches for quantification of bioactive fatty acids and associated oxylipins (PL-eicosanoids).

**Table 1 nutrients-11-02974-t001:** Influence of fatty acids and their metabolites on lymphocyte functions. Source—Endogenous/supplement—the fatty acid was supplemented in an in vivo study or enriched in formulations in vitro. Synthetic—fatty acid was used in in-vitro studies. PLA2—phospholipase A2, PGE2—prostaglandin E2.

Lipid	Source	Immune Cell	Function	Ref.
Fatty acids				
FA 20:4, FA 20:5, FA 22:6	Endogenous, supplement	Neutrophil	Adherence to endothelia (CD11a and CD 11b)	[32]
FA 18:3 n-3	Supplement	Alveolar macrophages	Increased phagocytosis, Increased TNF-αproduction	[20,33]
FA 18:3 n-3	Oral	T-cell	Suppress T cell proliferation	[70]
FA 20:4	PLA2-II mediated release of arachidonic acid (only release no metabolism)	Neutrophil	Increased mac-1 (CD-11b/CD18) expression	[59]
FA 18:0, 18:2, 18:3, 20:4	Endogenous	Macrophages and hepatocytes	Ligand binding activators of PPAR-α, PPAR-γ	[41]
FA 18:2 n-6	Dietary source	Dendritic cells	Reduced infiltration of LN and activation of T-cell. Reduced IL-12 increased IL -10	[71,72]
FA 20:5	Synthetic	Mast Cells	Decreased activation	[73]
FA 22:6 n-3	Synthetic	Dendritic cells	Increased IL-12 Reduced IL-6 and IL-10	[74]
FA Metabolites				
PGE2	Endogenous	Lymphocytes	Inhibitor T_H_1 response (IL-12)	[21]
Leukotriene B4	Endogenous, supplement	Neutrophil	Adherence to endothelia (CD11a and CD 11b)	[32]
Inflammasome Palmitic acid (C16:0) Oleic acid (C 18:1)	Supplement Supplement and dietary sources	NLRP3 inflammasome NLRP3 inflammasome	Increased IL-1β, IL-18 Decreased IL-1β, TNF-α, IL-6	[44,45] [43]

**Table 2 nutrients-11-02974-t002:** Clinical studies involving polyunsaturated fatty acids.

Study Design	Lipids	Study Endpoints	Results	Ref.
Double blind RCT study	Omega-3 long-chain polyunsaturated fatty acids	Allergic symptoms in children from mothers supplemented with 2.7 g omega-3 LCPUFA daily	Fewer allergies in children whose mothers received high omega-3 LCPUFA supplement.	[39]
REDUCE-IT study—double blind RCT study	Eicosapentaenoic acid	Cardiovascular death	2 g of EPA twice daily reduce risk of ischemic events.	[82]
Double blind RCT study	Eicosapentaenoic acid	Reduction of depressive symptoms	Omega-3 supplementation benefit patients with major depressive episode without comorbid anxiety disorder.	[83]
PREDIMED study random subsample—parallel-group randomized trial	Mediterranean diet	Cardiovascular events	Incidence of cardiovascular events was lower in patients receiving Mediterranean diet supplemented with extra-virgin oil or nuts.	[84]
PREDIMED study random subsample—parallel-group randomized trial	Mediterranean diet	Effect of HDL particles on reverse cholesterol transport	The diet increased cholesterol efflux, decreased cholesteryl ester transfer protein activity and increased HDL ability to esterify cholesterol.	[37]
Double blind RCT study	Fish oil n-3-PUFA	Muscle strength and average isokinetic power	n-3 PUFA therapy slows muscle mass decline and function in older adults	[85]
Double blind RCT study	Fish oil n-3-PUFA	Response of lysophospholipids to obesity	Obesity impact lysophospholipid metabolism abolishing its sensitivity to n-2 PUFA.	[86]
Compassionate protocol	Fish oil-based lipid emulsion	Resolution of cholestasis (plasma conjugated bilirubin <2 mg/dL)	All survival demonstrated resolution of cholestasis, compared with only 10% of non-surviving.	[87]
Double blind RCT study	EPA-DHA intake	Creatinine–cystatin C-based GFR	Long term supplementation with 400 mg/d of EPA-DHA provides slower kidney function decline in CKD patients.	[88]
Open-label randomized study	EPA-DHA intake	Cumulative rate of all-cause death, non-fatal myocardial infarction, and non-fatal stroke	Treatment significantly lowered risk of death and cardiovascular death.	[89]

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
