# Peer review of "Dietary Bioactive Fatty Acids as Modulators of Immune Function: Implications on Human Health"

_nutrients, 2019, doi:10.3390/nu11122974_

Round 1

Reviewer 1 Report

The authors have presented a review on the topic of bioactive immune-modulators and the impact on human health.

1-The concept of dietary fatty acids in regulating immune cells is emerging, but there are very few clinical trials that led to similar conclusions of in vitro and ex vivo studies of isolated lipids. Please discuss.

2-Since this is a review related to human health, I suggest the authors add a table listing the clinical trials that show evidence that fatty acids are able to benefit human health and longitudinal trials with the disease.

3-While the review is well-written and informative, it would be beneficial to add a box with GPCR receptors that are affected and their cognate lipid ligands, intermediate compounds in health and disease

Minor Revisions:

line 44: "an Increase" - an increase? line 45: "species" - genus? line 46: "health in humans" - health and disease? lines 70-71 "modulate signaling" -how? line 110: "Reviewed in 1" - figure 1? line 116: "Lekotrienes(LTx4)"-  Leukotrienes already spelled out, please emphasize on the numbers of the isoforms lines 126, 131: "incorporation" - repetition? line 135: "TLR" - plural? line 303: ".[77]." - remove the period?

Author Response

1-The concept of dietary fatty acids in regulating immune cells is emerging, but there are very few clinical trials that led to similar conclusions of in vitro and ex vivo studies of isolated lipids. Please discuss.

Response: Thank you for the thoughtful comment, we had added discussions on this topic in Section 2.3 of  the discussion.

2-Since this is a review related to human health, I suggest the authors add a table listing the clinical trials that show evidence that fatty acids are able to benefit human health and longitudinal trials with the disease.

Response: We have added Table 2 to point to clinical studies that are important in this field of research.

3-While the review is well-written and informative, it would be beneficial to add a box with GPCR receptors that are affected and their cognate lipid ligands, intermediate compounds in health and disease

 Response: Thank you for the thoughtful comment. While we understand that this is an interesting area of research, it is difficult to summarize this field within the focus of this review. We have now included references to a few comprehensive reviews on the GPCR related to free fatty acids.

Minor Revisions:

line 44: "an Increase" - an increase? line 45: "species" - genus? line 46: "health in humans" - health and disease? lines 70-71 "modulate signaling" -how? line 110: "Reviewed in 1" - figure 1? line 116: "Lekotrienes(LTx4)"-  Leukotrienes already spelled out, please emphasize on the numbers of the isoforms lines 126, 131: "incorporation" - repetition? line 135: "TLR" - plural? line 303: ".[77]." - remove the period?

Response: Thank you, we have made the necessary corrections.

Reviewer 2 Report

In this review, the authors detail the relationship between dietary fatty acids and immune function. This is an interesting topic, and one that is gaining interest within the scientific community, particularly as our ability to measure and analyse lipids improve. I think this review will be well received and will be of interest to your readers. The review, for the most part is well written, and addresses the topic in detail with a good grasp on the previous literature. Based on this I would recommend that this be published, but with some minor revision.

My chief concern is the in depth review of the current methodologies employed to measure lipids. I find this to be unnecessary and off topic, and think that this does not need to reviewed in depth in regards to the topic of this review (lines 80-94 and section 2.4). It is my belief that this review would be improved if this was removed and this would allow more room for additional review of the topic in question. Namely it would be interesting to elucidate the role of fatty acids in macrophage function or other aspect of immunity such as haematopoiesis.

Another minor comment would be to reword lines 39-44, as I found this was not very clear.

Author Response

In this review, the authors detail the relationship between dietary fatty acids and immune function. This is an interesting topic, and one that is gaining interest within the scientific community, particularly as our ability to measure and analyse lipids improve. I think this review will be well received and will be of interest to your readers. The review, for the most part is well written, and addresses the topic in detail with a good grasp on the previous literature. Based on this I would recommend that this be published, but with some minor revision.

My chief concern is the in depth review of the current methodologies employed to measure lipids. I find this to be unnecessary and off topic, and think that this does not need to reviewed in depth in regards to the topic of this review (lines 80-94 and section 2.4)

Response: Thank you for your comment. We understand your concern with the methods sections. It has now been significantly shortened to be within the context of the focus of the review. With great respect to the reviewer however, it is our belief that a brief introduction about the methods currently used to quantify bioactive lipids and oxidized phospholipids is important to describe the technologies available to study the metabolism of bioactive lipids. As such, we have included a much shortened version of this section and also have added an updated Figure 3 reflecting the relevant changes.

It is my belief that this review would be improved if this was removed and this would allow more room for additional review of the topic in question. Namely it would be interesting to elucidate the role of fatty acids in macrophage function or other aspect of immunity such as hematopoiesis.

Response: Thank you. We have used the space gained by shortening the lipid analytical methodologies and used that space to add a discussion on M1/M2 polarization and the influence of lipid composition and eicosanoid lipid metabolism.

Another minor comment would be to reword lines 39-44, as I found this was not very clear.

Response: The section has been revised to improve clarity.